# Reversed asymmetric warming of sub-diurnal temperature over land during recent decades

Ziqian Zhong [1,2], Bin He [1] ✉, Hans W. Chen [2], Deliang Chen [3], Tianjun Zhou [4], Wenjie Dong[5], Cunde Xiao[6], Shang-ping Xie [7], Xiangzhou Song [8], Lanlan Guo[1], Ruiqiang Ding [6], Lixia Zhang[4], Ling Huang[9], Wenping Yuan [5], Xingming Hao[10], Duoying Ji [1] & Xiang Zhao[11]

In the latter half of the twentieth century, a significant climate phenomenon "diurnal asymmetric warming" emerged, wherein global land surface temperatures increased more rapidly during the night than during the day. However, recent episodes of global brightening and regional droughts and heatwaves have brought notable alterations to this asymmetric warming trend. Here, we re-evaluate sub-diurnal temperature patterns, revealing a substantial increase in the warming rates of daily maximum temperatures ($T_{max}$), while daily minimum temperatures have remained relatively stable. This shift has resulted in a reversal of the diurnal warming trend, expanding the diurnal temperature range over recent decades. The intensified $T_{max}$ warming is attributed to a widespread reduction in cloud cover, which has led to increased solar irradiance at the surface. Our findings underscore the urgent need for enhanced scrutiny of recent temperature trends and their implications for the wider earth system.

The surface air temperature (SAT) is a commonly used measure of land surface climate change due to its ability to represent terrestrial energy exchange with reasonable accuracy[1,2]. In addition to daily average temperatures, the diurnal temperature range (DTR), defined as the difference between the daily maximum temperature ($T_{max}$) and daily minimum temperature ($T_{min}$), provides useful information about the climate[3,4]. Changes in DTR have received considerable attention because it is closely linked with crop yields[5–7], plant growth[8–11], animal wellbeing[12,13] and human health[14–16]. Existing studies found that the

surface warming since the 1950s has been associated with larger increases in $T_{min}$ than in $T_{max}$, i.e., decreases in DTR, which is commonly known as night warming or asymmetric warming[17–20].

Changes in DTR are complex; the changes are subject to many factors, including cloud cover[21–23], solar radiation[24,25], aerosols[26], precipitation[27,28], planetary boundary layer height[29], land use change[30,31] and deforestation[32,33]. For example, an increase in total cloud cover reduces DTR due to a decrease in daytime surface insolation and an increase in night-time downwards longwave radiation[34].

[1]State Key Laboratory of Earth Surface Processes and Resource Ecology, Faculty of Geographical Science, Beijing Normal University, Beijing 100875, China. [2]Department of Space, Earth and Environment, Division of Geoscience and Remote Sensing, Chalmers University of Technology, SE-412 96 Gothenburg, Sweden. [3]Regional Climate Group, Department of Earth Sciences, University of Gothenburg, S-40530 Gothenburg, Sweden. [4]Institute of Atmospheric Physics, Chinese Academy of Sciences, Beijing 100029, China. [5]School of Atmospheric Sciences, Sun Yat-Sen University, Guangzhou 510275, China. [6]State Key Laboratory of Earth Surface Processes and Resource Ecology, Beijing Normal University, Beijing 100875, China. [7]Scripps Institution of Oceanography, University of California San Diego, La Jolla, CA 92039, USA. [8]Key Laboratory of Marine Hazards Forecasting, Ministry of Natural Resources (MNR), Hohai University, Nanjing 210024, China. [9]College of Urban and Environmental Sciences, Peking University, Beijing 100871, China. [10]State Key Laboratory of Desert and Oasis Ecology, Xinjiang Institute of Ecology and Geography, Chinese Academy of Sciences, Urumqi 830011, China. [11]State Key Laboratory of Remote Sensing Science, Beijing Normal University, Beijing 100875, China. ✉e-mail: hebin@bnu.edu.cn

Increases in precipitation and soil moisture can reduce $T_{max}$ and therefore DTR through increased evaporative cooling[35]. However, considering known changes in these processes, the higher warming rate of $T_{min}$ than that of $T_{max}$ is seemingly inconsistent with two recent phenomena. One contradiction is the increase in surface solar radiation that has occurred since the late 1980s, which is referred to as brightening after dimming[36–38]. Solar radiation affects $T_{max}$ more than $T_{min}$[26], thus, brightening should contribute to further warming of $T_{max}$. The second contradiction is the increased occurrence of drought events and heatwaves, especially in spring and summer[39–41]. This phenomenon indicates that the cooling effect of soil moisture may have been weakened, which should lead to a faster increase in $T_{max}$. Motivated by these different changes, we re-evaluated the warming rates of $T_{max}$ and $T_{min}$ over the period of 1961–2020 and investigated the causes behind the associated changes in DTR.

## Results

### Observed reversing asymmetric warming

With changes in recent years included, we evaluated the surface warming rates of $T_{max}$ and $T_{min}$ using two station observation-based datasets from Berkeley Earth Surface Temperatures (BEST)[42] and the Climatic Research Unit Time-Series version 4.07 (CRU TS)[43]. Both datasets were gridded and gap-filled over the land masses, and BEST was used to detect global surface temperature changes in the Intergovernmental Panel on Climate Change (IPCC) sixth assessment report[44]. As shown in Fig. 1a, b, the trends in the global area-weighted average of $T_{max}$ calculated using a 30-year moving window increased faster than that of $T_{min}$ during 1961–2020. The warming rate in global average $T_{max}$ reached the warming rate in $T_{min}$ in recent decades, with an earlier surpassing moment in BEST than in CRU TS. In the last 30-year window (1991–2020), both datasets exhibit a slight (CRU TS dataset) or substantially more pronounced (BEST dataset) rise in the global average of $T_{max}$ compared to that of $T_{min}$. Spatially, a stronger warming rate of $T_{max}$ was found in almost 25% of the land area in the earlier time window (1961–1990); the area expanded rapidly, and in the recent time window (1991–2020) the area of stronger warming rate of $T_{max}$ was found in at least half of the area (approximately 52% in CRU TS and 70% in BEST, Fig. 1e, f, with consistent results between the two datasets in 61% of the total land area). Observations from stations based on Global Surface Summary of the Day (GSOD) support the finding of a greater increase in $T_{max}$ than in $T_{min}$ from 1991–2020 (Fig. 2). Analyzes of all observed monitoring sites indicate that 63% of the sites exhibited an overall upward trend in DTR, while more than one-third (35%) of all sites displayed a statistically significant increase in DTR during 1991–2020. Spatially, a widespread decreasing trend in DTR was detected (81% and 76% land area fraction in the BEST and CRU TS temperature datasets, respectively) for the period 1961–1990, with the exceptions of southern Africa, Southern Europe, and some regions in Northern America (Fig. 1c, d). However, during the recent decades of 1991–2020, both temperature datasets show a consistent increase in DTR over more than half of the land area, in particular over the western United States, southern Europe, West Africa, inner East Asia, and Australia.

Our analyzes reveal that the reversal of asymmetric warming was more pronounced in the BEST dataset compared with CRU TS. We assessed the accuracy of these two sets of gridded DTR data by comparing them against GSOD station DTR data (see Methods). The findings revealed a significantly stronger correlation between BEST's DTR data and the GSOD station dataset (one-tailed $t$-test, $p < 0.001$) compared to the correlation between CRU TS's DTR data and the GSOD station dataset (Supplementary Fig. 1). Thus, in the subsequent analysis we used the BEST dataset to study annual and seasonal trends in $T_{max}$, $T_{min}$ and DTR for the periods 1961–1990 and 1991–2020.

A significant decline in the global average of DTR (−0.08 °C decade$^{-1}$, $p < 0.05$) was found during 1961–1990. For the period 1991–2020, the globally averaged DTR increased at a rate of 0.06 °C decade$^{-1}$ ($p < 0.05$) (Supplementary Fig. 2c). This reversal in DTR occurred mainly due to a marked increase in $T_{max}$ for the latter period (0.35 °C year$^{-1}$ decade$^{-1}$, $p < 0.05$ during 1991–2020 vs. 0.13 °C year$^{-1}$ decade$^{-1}$, $p < 0.05$ during 1961–1990, Supplementary Fig. 2a, b). Seasonally, the largest decline in DTR was detected in winter during 1961–1990, which is consistent with previous findings[35]. Significant increases ($p < 0.05$) in DTR were detected in spring, summer and winter, which were mainly caused by a significant increase in $T_{max}$ (Supplementary Fig. 2a). All these evidences point to a reversing asymmetric warming over land in recent decades.

### Potential mechanisms behind reversing asymmetric warming

Earlier studies suggested that the recent global warming is mainly forced by greenhouse gases[45], while changes in DTR are influenced largely by cloud cover[23,46]. In addition, aerosols and soil moisture may also have affected the variability of DTR[47,48]. These potential drivers of DTR exhibit a high degree of correlation. For instance, the phenomenon of aerosol-cloud interactions, such as the influence of aerosols on cloud albedo[49] and lifetime[50], has gained growing attention[51–53]. Moreover, clear associations can be observed between cloud cover and soil moisture, as increasing cloud cover is associated with enhanced precipitation[54], which subsequently leads to soil wetting. The intricate interactions among these factors highlight the presence of high multi-collinearity when performing regression analysis with them as independent variables, posing challenges in discerning the dominant drivers of DTR change. To address the impact of multi-collinearity and accurately identify relationships, we conducted ridge regression analyzes[55] at individual grid points. This analysis (Methods) employed as independent variables monthly total cloud cover from the fifth-generation ECMWF reanalysis (ERA5)[56] dataset, aerosol optical depth from Modern-Era Retrospective analysis for Research and Applications, version 2 (MERRA-2)[57], and soil moisture from the Global Land Evaporation Amsterdam Model (GLEAM)[58] dataset, while DTR was calculated from the BEST dataset as the dependent variable spanning 1981 to 2020. Ridge regression is a linear regression method with regularization that can effectively address the issue of multi-collinearity. By introducing a penalty term in the cost function that discourages overly large parameter values, it improves upon the ordinary least squares regression model in scenarios where there exist strong correlations among independent variables. An evaluation of the ridge regression (Methods) shows that the ridge regression model can capture the majority of the explained variance in DTR (Supplementary Fig. 3a), except in certain regions in particularly Africa, South America, and Northern Hemisphere high-latitudes.

The analysis using ridge regression suggests a worldwide negative response of DTR to changes in total cloud cover and soil moisture (Supplementary Fig. 3b, d), with generally stronger negative impacts of cloud cover than soil moisture. This result aligns with previous findings[21,23]. Negative responses of DTR to aerosol concentrations are detected in Western Africa, the Arabian Peninsula, India and southern China (Supplementary Fig. 3c). To make a quantitative comparison, Fig. 3 shows a cyan-magenta-yellow (CMY) composite map (see Methods) of the relative contributions of total cloud cover, aerosol and soil moisture to DTR. Generally, changes in DTR during 1981–2020 were dominated by cloud cover variations over 83.5% of the global land area; this phenomenon is widely detected in most of North America, southern South America, Europe, southern Africa, Central Asia, and East Asia. Moreover, the dominant effect of cloud cover on DTR over land is further confirmed when using an independent total cloud cover dataset obtained from the Moderate Resolution Imaging Spectroradiometer (MODIS) satellite during 2003–2020 (84.4% of the total land area; Supplementary Fig. 4).

The ridge regression analysis assumes linear relationships and may overlook the non-linear associations among cloud, aerosol,

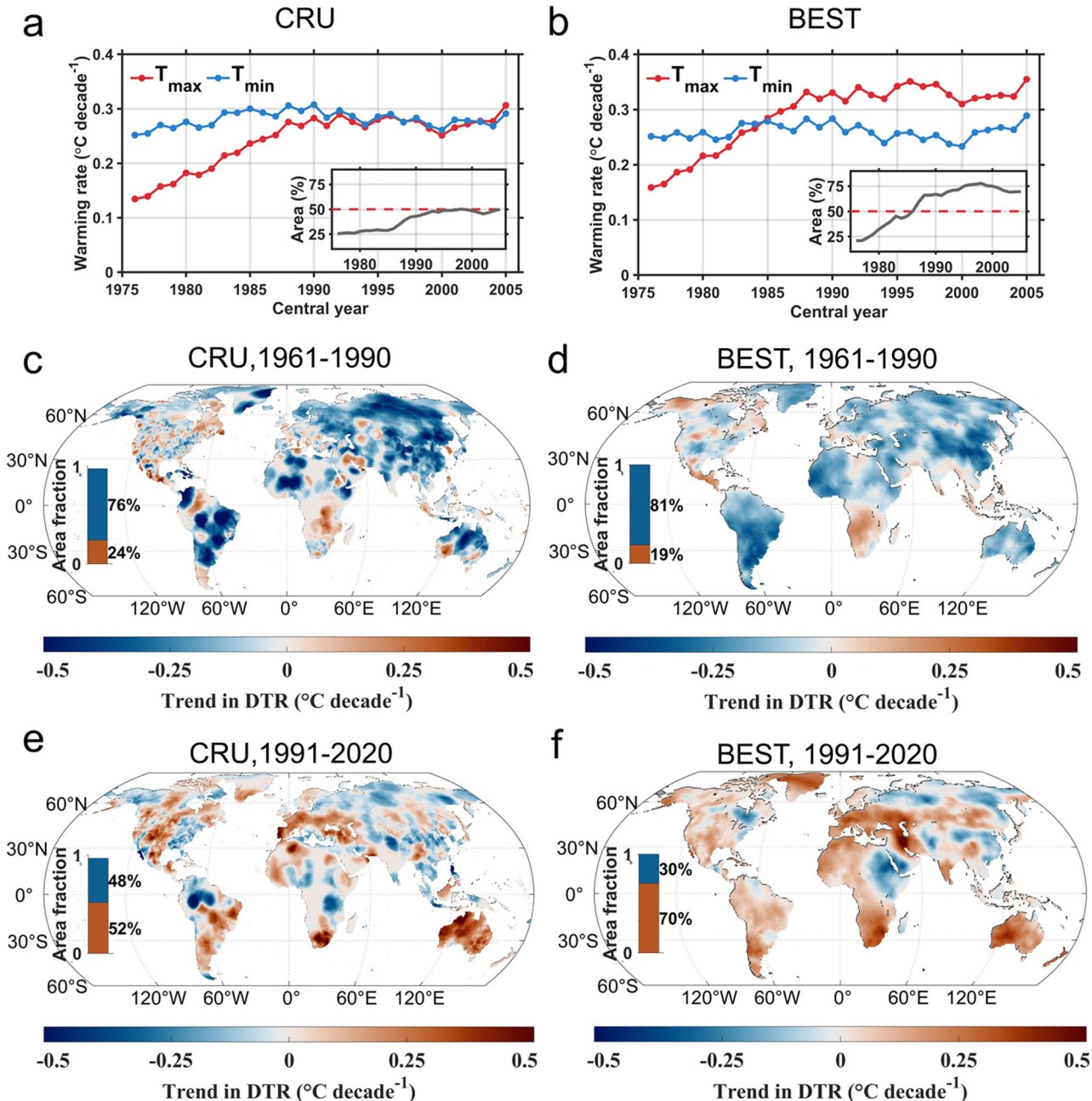

**Fig. 1 | Trends in the daily maximum temperature ($T_{max}$), daily minimum temperature ($T_{min}$) and diurnal temperature range (DTR). a, b** Global area-weighted average warming rates derived from the CRU TS (**a**) and BEST (**b**) datasets in $T_{max}$ (red) and $T_{min}$ (blue). The trends were calculated using a 30-year moving window over 1961–2020. The *x*-axis shows the central year (rounding down) of the moving window. The inset shows the area fraction over land (%) with faster warming rates of $T_{max}$ than $T_{min}$. **c**–**f** Spatial distribution of the trend in DTR in CRU TS during 1961–1990 (**c**) and 1991–2020 (**e**), and in BEST during 1961–1990 (**d**) and 1991–2020 (**f**).

soil moisture, and DTR. To address this limitation, we repeated the regression analysis using the Random Forest algorithm[59] (Methods). Random Forest is a machine learning technique based on decision trees and can capture non-linear relationships. The results from the Random Forest regression consistently corroborate our prior findings, indicating that total cloud cover is indeed the dominant driving force behind DTR fluctuations in terrestrial regions, encompassing a substantial proportion (79.1%) of the land area (Supplementary Fig. 5). DTR is significantly negatively correlated with total cloud cover in most regions of the terrestrial land surface, suggesting that DTR is greatly influenced by cloud cover.

To further elucidate the influence of different environment variables on DTR, we analyzed the changes of DTR, total cloud cover, aerosol optical depth and soil moisture over the global land area during recent decades (Supplementary Fig. 6). During the 1960s to the early 1970s, there was a strong and significant upward trend in total cloud cover, which later stabilized. However, a remarkable downward trend emerged from the mid-1980s onwards. This decline in cloud cover corresponded with an increase in DTR during the same period. Between 1981 and 2020, a significant ($p < 0.05$) negative correlation was found between the annual average global surface total cloud cover and DTR. The mean bootstrapped partial correlation coefficient (−0.47; −0.54 to −0.41, 95% confidence interval (CI)) between these two

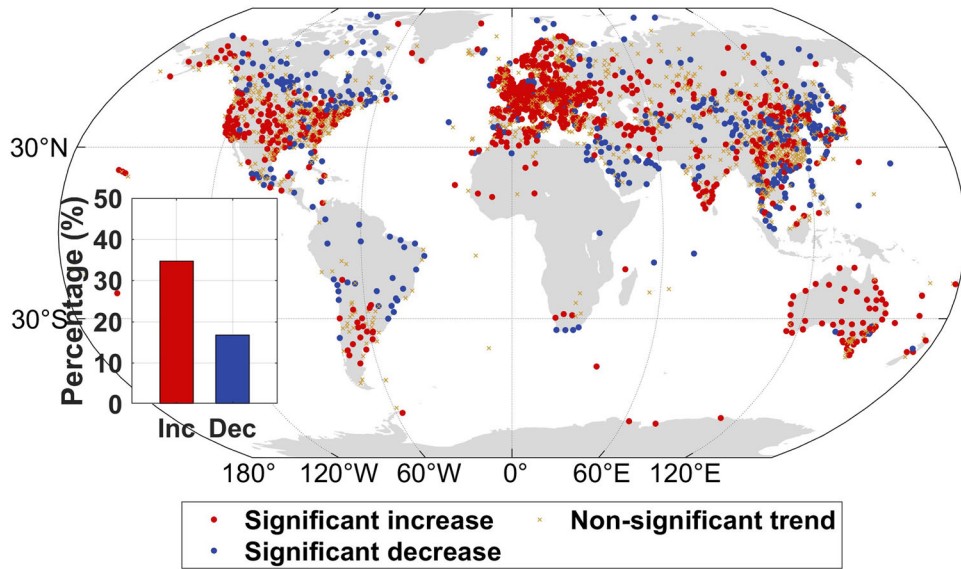

**Fig. 2 | Spatial distribution of the trend in diurnal temperature range (DTR) in Global Surface Summary of the Day (GSOD) during 1991–2020.** The insets in the figure depict the percentage of sites showing a significant increasing (Inc; $p < 0.05$; red) and significant decreasing (Dec; $p < 0.05$; blue) trend in DTR.

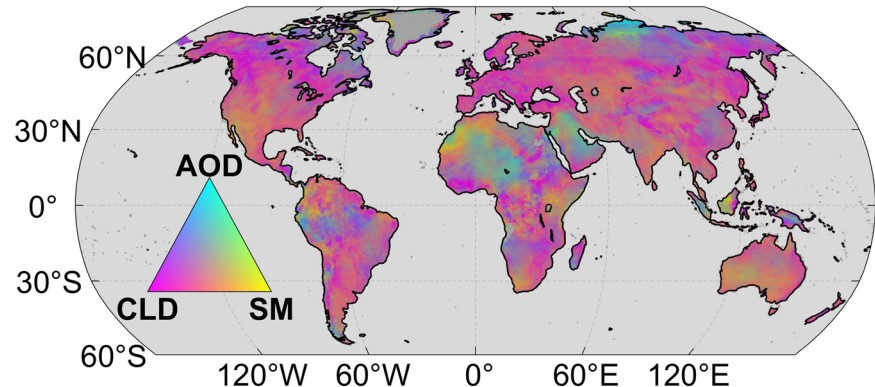

**Fig. 3 | Cyan-magenta-yellow (CMY) composite of diurnal temperature range (DTR) sensitivity.** The contributions of total cloud cover (CLD; magenta), aerosol optical depth (AOD; cyan) and soil moisture (SM; yellow) to DTR changes during 1981–2020. The color of the composite was determined by the relative contribution from the magnitude of the ridge regression coefficients. Only the grid cells with the regression result that passed the test of significance ($p < 0.05$) in the training set are shown. The CLD was from the fifth-generation ECMWF reanalysis (ERA5) dataset, AOD was from Modern-Era Retrospective analysis for Research and Applications, version 2 (MERRA-2), and SM was from the Global Land Evaporation Amsterdam Model (GLEAM) dataset.

variables was higher than that of the mean correlation between DTR and soil moisture (−0.31; −0.39 to −0.23, 95% CI), as well as the correlation between DTR and aerosol optical depth (−0.18; −0.28 to −0.07, 95% CI).

Cloud cover can influence DTR through two primary mechanisms. On the one hand, diminished cloud cover leads to greater daytime solar radiation, resulting in elevated daytime maximum temperatures and, consequently, an expanded DTR. Conversely, reduced cloud cover leads to decreased night-time outgoing radiation at the surface, causing lower minimum temperatures and further increasing DTR[23,46]. To discern the predominant effect, we calculated the partial correlations between both cloud cover and $T_{max}$ and cloud cover and $T_{min}$ (Supplementary Fig. 7). The results indicate that, overall, the prevalent association is a negative correlation between cloud cover and maximum temperatures. This suggests that decreased cloud cover has led to an increased DTR in recent decades, primarily due to the associated increase in incoming solar radiation, while the impact of night-time cooling has played a secondary role.

To reveal the spatial patterns of cloud and radiation changes, we compared the trends in total cloud cover and solar radiation during the earlier decades (1961–1990) and the recent decades (1991–2020) (Fig. 4). Specifically, a global (over 69% of land) increase in cloud cover corresponded to a decline in solar radiation at the surface over almost three-quarters (74%) of the land area for the period 1961–1990; the dimming has disappeared since the 1990s, and increases in solar radiation were detected in over 63% of global land, which is partly due to an extensive (over 66% of land) decrease in cloud cover. Over the period 1961–1990, central North America, southern South America, the Mediterranean, and Australia saw weak decreases in solar radiation, with the local DTR narrowing or experiencing little change. However, during 1991–2020, a substantial increasing trend in solar incident radiation was detected over central North America, southern South America the Mediterranean, and Australia, corresponding to a local general increase of DTR.

To further elucidate the relationship between incident solar radiation and DTR, we conducted a partial correlation analysis between DTR and incident shortwave radiation extracted from ERA5 (1961–2020), CERES (2001–2020), and MERRA-2 (1981–2020), while adjusting for the influence of soil moisture (Supplementary Fig. 8). The results show a significant positive correlation between

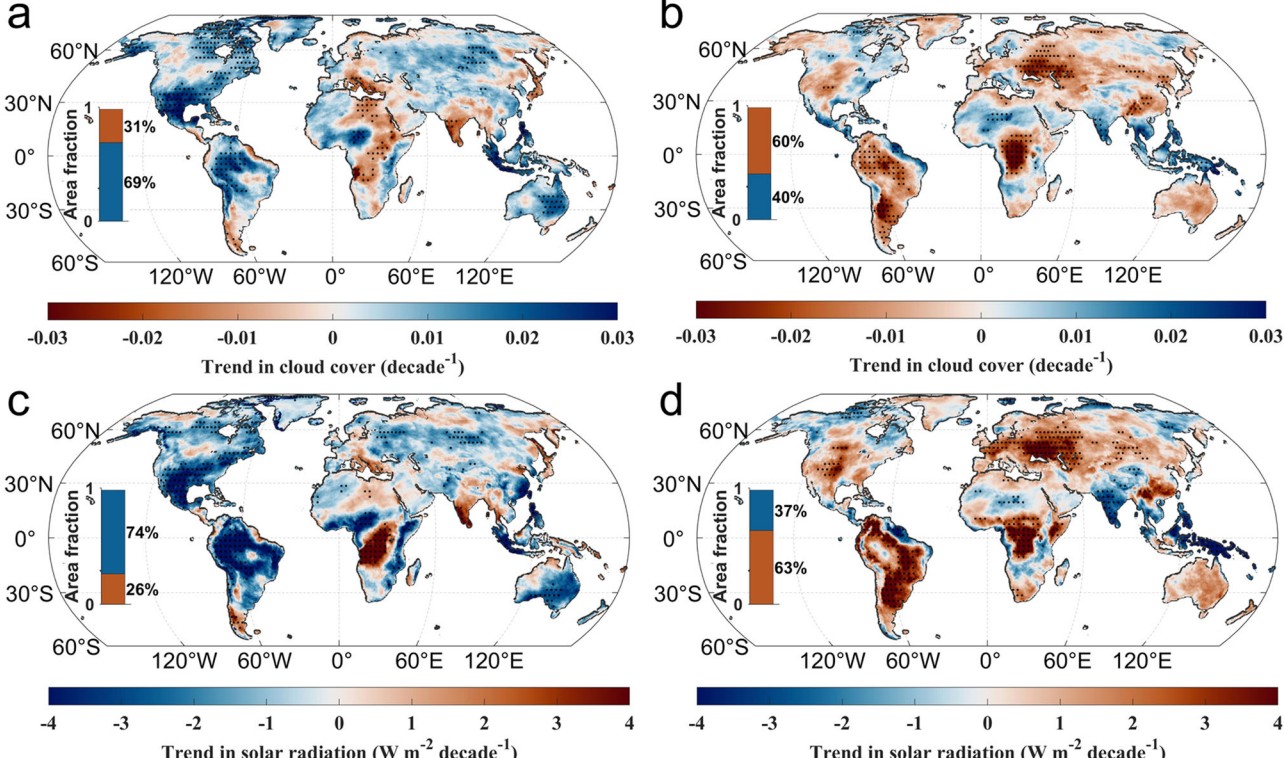

**Fig. 4 | Comparisons of trends in total cloud cover and solar radiation between two periods. a**, **b** The spatial distribution of trends in total cloud cover during 1961–1990 (**a**) and 1991–2020 (**b**). **c**, **d** The spatial distribution of trends in solar radiation during 1961–1990 (**c**) and 1991–2020 (**d**). The black dots mark the areas where trends are significant at the $p < 0.05$ level. The total cloud cover and solar radiation were from the fifth-generation ECMWF reanalysis (ERA5) dataset.

DTR and surface solar radiation over most land areas of the world in all four datasets, confirming a credible positive influence of solar radiation on DTR, consistent with previous studies[24,25]. The above results indicate that the recent global decrease in cloud cover[60–62] has increased incoming solar radiation at the surface, resulting in a greater increase in $T_{max}$ than in $T_{min}$, ultimately expanding DTR.

Our analyzes show that the reversed asymmetric warming was mainly driven by changes in cloud cover which led to enhanced solar radiation. Further, the influence of soil moisture or aerosols on DTR appears to be particularly noticeable in certain regions. Although the influence of soil moisture on DTR may not be as substantial as that of cloud cover, it shows relatively extensive spatial distributions across terrestrial surfaces on a global scale (Supplementary Fig. 3d). In those regions, decreasing soil moisture may have contributed to the acceleration of $T_{max}$ increases and thus decreasing DTR, possibly due to less effective daytime evaporation cooling on air temperature during dry conditions. Additionally, the increase in DTR in the Mediterranean over the past three decades may be related to a decrease in aerosol concentrations (Supplementary Fig. 9b), which is likely linked to reductions in local aerosol and precursor emissions[63]. However, due to the little change in global drought[64] and the relatively short-lived influence of aerosol on DTR[65], the overall impact of soil moisture and aerosols on the global asymmetric warming pattern was limited. In addition to these three environmental factors, changes in DTR may be closely associated with land use/land cover change (LULCC). Given the complexity of LULCC's effects, our study specifically focuses on exploring one potential impact: changes in albedo. However, we found that there was no significant positive or negative correlation between changes in albedo and DTR across the majority of land areas (Supplementary Fig. S10). This finding indicates a limited influence of LULCC on global asymmetric warming.

## Discussion

The reversing asymmetric warming is supported by regional increasing DTR trends in Europe[66], Central Asia[67], South India[68] and Australia[69] during recent decades. Globally, it was reported that most of the global-mean DTR decrease occurred between 1960 and 1980. After that period, globally averaged DTR exhibited little change from 1979 to 2012[70]. Here we found that global DTR has reversed from decreasing to increasing in the recent three decades based on two observation-based datasets and station observations. While there are differences between the two gridded temperature datasets, particularly in South America and Africa where the observational coverage is limited[71,72], the increasing DTR in Europe, Australia, and Northern America is robust due to the dense observations in these regions. It is worth noting that the large number of land stations (approximately 39,000 records) and the diverse range of sources (8 sources) integrated within the BEST temperature dataset could potentially explain its higher correlation with the observed temperatures from the stations in the GSOD dataset, as compared to the CRU TS temperature dataset. In addition to disparities in station selection, variations in gridding and interpolation methodologies[69], as well as the implementation of quality assurance procedures[73], may also contribute to the differences in detected regional DTR trends between these two datasets. With that said, irrespective of the dataset employed, it is evident that more than half of the global land surface has displayed a discernible increasing trend in DTR over the past three decades. This finding signals a fundamental shift in the pattern of asymmetric warming.

The results of the ridge regression analysis in this study suggest that statistical models incorporating total cloud cover, aerosol optical depth, and soil moisture as independent variables can effectively capture the variations in DTR across most land areas. The model fit is relatively low in regions such as South America and Africa (Supplementary Fig. 3a) where the observational coverage is limited. These regions are also where the CRU TS and BEST gridded temperature

datasets show the largest discrepancies in DTR trends in the later period (1991–2020), which shows that there are considerable uncertainties in the estimated DTRs in regions with few observations. Additionally, in this study, we did not include other potentially relevant factors such as precipitation[28,74], atmospheric water vapor[23,75], vegetation[76], LULCC[45,77,78] and regional atmospheric circulation[79]. This omission may introduce further uncertainty. Increased precipitation, for example, is closely linked to decreased DTR, which can be attributed to the strong association between precipitation, cloud cover, atmospheric water vapor, radiation, and soil moisture[29,74]. Future studies are needed to disentangle the effects of these highly correlated variables on DTR. Prior research conducted in specific regions, such as China and India[77,78], has demonstrated that LULCC exerts a notable influence on DTR. In light of the intricate nature of LULCC's effects, our study specifically focuses on exploring the potential impact of changes in albedo. However, we found that there is a relatively weak correlation between DTR and albedo across the majority of land areas. This could be attributed to the fact that LULCC encompasses other processes, such as vegetation dynamics[76,80,81]. Further studies are needed to gain a comprehensive understanding of the entire scenario.

We found that the reversed asymmetric warming is closely linked with changes in solar radiation associated with total cloud cover. This finding offers fresh insights into and a different perspective on global climate change in recent decades. Given that clouds may continue to have a positive feedback on global warming through radiative fluxes[82,83], this radiation-induced phenomenon, in which the rising rate of $T_{max}$ exceeds that of $T_{min}$, may persist and potentially intensify in the future. Therefore, more attention needs to be paid to this asymmetric warming phenomenon from the perspective of tackling the ongoing challenges posed by global warming.

## Methods

### Data

The $T_{max}$ and $T_{min}$ data utilized in this study were obtained from the Climatic Research Unit Time-Series version 4.07 (CRU TS)[43] and Berkeley Earth Surface Temperatures (BEST) datasets[42]. The CRU TS temperature dataset is derived from a blend of data sources including weather station records, ship logs, and more recent satellite observations. This dataset undergoes meticulous calibration to account for biases and variations in measurement methods, making it an extensively used resource in climate research for documenting long-term temperature trends and variations. It provides spatiotemporal resolutions of $0.5° × 0.5°$ on a monthly basis, covering the period from 1901 to 2022.

The BEST dataset encompasses a larger sample of approximately 39,000 records. It employs advanced statistical techniques to quantify and adjust measurement biases, ensuring a highly accurate representation of global temperature trends. This dataset offers spatiotemporal resolutions of $1° × 1°$ on a monthly basis, spanning from 1850 to the present.

In situ $T_{max}$ and $T_{min}$ records were acquired from the Global Surface Summary of the Day (GSOD) database, downloaded in September 2021 from https://data.nodc.noaa.gov/cgi-bin/iso?id=gov.noaa.ncdc: C00516. The GSOD dataset originates from the Integrated Surface Hourly (ISH) database developed by the US National Climatic Data Center. This dataset contains meteorological variables, including temperature, precipitation, and wind speed, from over 9000 weather stations. The original observations undergo rigorous quality control procedures to ensure their accuracy[84]. Specifically, we selected daily maximum and minimum temperature data with relatively complete records from all stations covering the period 1991 to 2020. We employed stringent selection criteria to exclude incomplete data series, only including stations with missing values not exceeding one year of the analysis period (1991 to 2020) and with complete records for all 12 months. Furthermore, we required each monthly value to be

derived from at least 15 days of data. Consequently, our trend analysis encompassed a total of 2557 stations.

Monthly data on total cloud cover were obtained from the fifth-generation ECMWF reanalysis (ERA5) dataset[56] on a $0.25° × 0.25°$ regular latitude–longitude grid (the native resolution of ERA5 is about 31 km). In addition, monthly total cloud cover was obtained from the MODIS MCD06COSP data at a spatial resolution of $1° × 1°$ after 2003. ERA5 total cloud cover was used for the main analyzes.

The monthly incident shortwave radiation data in all-sky conditions were obtained from the ERA5 dataset on a $0.25°$ grid, the Cloud and the Earth's Radiant Energy System energy balanced and filled edition 4.1 (CERES–EBAF)[85] dataset at a spatial resolution of $1° × 1°$ after March 2000, and the Modern-Era Retrospective analysis for Research and Applications, version 2 (MERRA-2)[57] dataset with a spatial resolution of $0.625° × 0.5°$ after 1980.

The monthly average aerosol optical depth and surface albedo at a spatial resolution of $0.625° × 0.5°$ was obtained from the MERRA-2 dataset beginning in 1980. The monthly surface soil moisture at a spatial resolution of $0.25°$ was obtained from the Global Land Evaporation Amsterdam Model (GLEAM) version 3.5 dataset[58], which is a global dataset spanning 40 years from 1981 to 2020 and based on satellite and reanalysis data. The cloud cover, radiation, aerosol optical depth, and soil moisture data sets were aggregated to a common $0.5°$ grid.

### Assessment of temperature-gridded data

We assessed the accuracy of the temperature gridded data using GSOD station temperature dataset. The evaluation period spanned from 1978 to 2020, during which the GSOD data records were relatively complete. Following a similar methodology used in analyzing station-based DTR trends, we selected a total of 2,058 stations with missing observations comprising less than 5% of the evaluation period for temperature-gridded data assessment. In the evaluation process, we extracted the time series of DTR from the gridded BEST and CRU TS datasets corresponding to each station's location. Daily data were averaged to obtain yearly data and missing values at these stations were filled using linear interpolation. We then computed Pearson correlation coefficients between the annual station observation series and the observed DTR sequence at each station, serving as an indicator of accuracy. The significance of the correlation coefficient differences between the DTR station data and the two sets of DTR gridded data was tested by the one-tailed Student's $t$-test.

### Seasonal analysis

To analyze DTR variations by season, we defined the seasons as follows: March, April, May for spring (autumn), June, July, August for summer (winter), September, October, November for autumn (spring), and December, January, February for winter (summer) in the Northern Hemisphere (Southern Hemisphere). Correspondingly, the annual average value of a variable in one year is defined as the average of 12 months from December in the preceding year to November of that year.

### Ridge regression

Ridge regression is a method for estimating coefficients in multiple linear regression models in scenarios characterized by high correlations among the independent variables. In ridge regression, a regularization term is introduced to the standard least squares objective function, aiding in stabilizing the estimated coefficients. This regularization term is controlled by a tuning parameter denoted as $\lambda$. The ridge regression objective function can be expressed as follows:

$$\beta^{\wedge} = \sum_{i=1}^{n} \left( y_i - \beta_0 - \sum \beta_i x_i \right)^2 + \lambda \sum \beta_i^2 \qquad (1)$$

where $\hat{\beta}$ represents the estimated regression coefficients, $y_i$ is the dependent variable, $\beta_0$ is the intercept term, and $\beta_i$ signifies the regression coefficient for the independent variable $x_i$. The tuning parameter $\lambda$ governs the extent of shrinkage applied to the coefficients. When $\lambda$ is set to zero, the regularization term exerts no effect, and ridge regression reduces to ordinary least squares regression. However, as $\lambda$ increases, the penalty term gains influence. The larger $\lambda$ becomes, the more pronounced is the shrinkage applied to the coefficients. Consequently, ridge regression effectively mitigates the impact of multi-collinearity.

Prior to performing the ridge regression analysis in this study, all-time series were detrended by subtracting the linear trend and transformed into z-scores by subtracting the monthly climatology means dividing by the monthly climatological standard deviations from 1981 to 2020. For validation, the z-scores were randomly divided into an 80% calibration dataset and a 20% validation dataset. The dataset in the training set was used to train the ridge regression model, while the dataset in the validation set was employed to assess the performance of the ridge regression model. Afterwards, all the datasets were merged to determine the ridge regression coefficients.

The ridge regression was performed on the variables in each grid cell, and the tuning parameter $\lambda$ was determined for each grid cell based on the Variance Inflation Factor (VIF) of the independent variables within each individual grid cell. VIF serves as a measure of multi-collinearity among the independent variables in the regression model. The following formula was used to calculate the VIF:

$$VIF_i = \frac{1}{1 - R_i^2} \qquad (2)$$

where $R_i^2$ denotes the coefficient of determination between the ith independent variable and all other independent variables. A higher VIF value indicates stronger multi-collinearity within the regression model's independent variables. Here, a VIF value less than 3 suggests an acceptable level of multi-collinearity[86].

Throughout the regression analysis process, the initial value of $\lambda$ was set to 0 and incremented by a step size of 0.01. As $\lambda$ increased, the degree of multi-collinearity decreased, subsequently resulting in a decline in the VIF value. The incrementation of $\lambda$ ceased when the VIF value dropped below 3, with this $\lambda$ value being determined as the tuning parameter at this grid point. The significance of the ridge regression analysis was assessed utilizing an F-test at a significance level of 0.05. The accuracy of the ridge regression model was evaluated by employing the coefficient of determination on the validation dataset.

## CMY composite

The color of the cyan-magenta-yellow (CMY) composite was determined by the relative contributions from the magnitudes of the ridge regression coefficients (R.c), which can be expressed as follows:

$$C = \frac{|R.c_{x1}|}{|R.c_{x1}| + |R.c_{x2}| + |R.c_{x3}|} \qquad (3)$$

$$M = \frac{|R.c_{x2}|}{|R.c_{x1}| + |R.c_{x2}| + |R.c_{x3}|} \qquad (4)$$

$$Y = \frac{|R.c_{x3}|}{|R.c_{x1}| + |R.c_{x2}| + |R.c_{x3}|} \qquad (5)$$

Here, $R.c_{x1}$, $R.c_{x2}$, $R.c_{x3}$ represent the R.c of DTR to variables $x_1$, $x_2$, and $x_3$, respectively. We denoted the relative contributions of variables $x_1$, $x_2$, and $x_3$ to DTR as C, M, and Y, respectively. These contributions

then served as the brightness values for the cyan, magenta, and yellow channels, respectively, culminating in the generation of a CMY image.

## Random Forest regression analysis

We also utilized the Random Forest algorithm for regression analysis[59]. The input to the Random Forest was standardized in the same way as for the ridge regression analysis, and the data was also randomly partitioned into an 80% calibration dataset and a 20% validation dataset for the validation. During the modeling process at individual grid cells, the Random Forest algorithm leveraged the provided sequences of independent variables and dependent variables in the calibration dataset to train 100 decision trees. Each decision tree independently predicted DTR values based on the given predictor variables. Moreover, we employed out-of-bag (OOB) prediction error estimation, an inherent capability of the Random Forest algorithm. Additionally, we assessed the importance of predictors by enabling the OOB Predictor Importance feature, providing insights into the relative contributions of total cloud cover, aerosol optical depth, and soil moisture in predicting DTR. Similar to the ridge regression coefficients, these Predictor Importance features were subsequently utilized in generating a CMY image.

## Partial correlation analysis and bootstrap method for assessing relationships

To determine the magnitude of the relationship between annual solar radiation and DTR, $T_{max}$ or $T_{min}$ at individual grid points, while controlling for the effect of soil moisture, we conducted a partial correlation analysis. Partial correlation analysis is a statistical technique used to assess the relationship between two variables while controlling for the influence of one or more additional variables. Here, both the variables involved in the correlation analysis and the variable being controlled for were specific to each individual grid cell. The significance of the partial correlations was evaluated at a threshold of $p < 0.05$.

The uncertainties of the partial correlations between global annual DTR and total cloud cover, soil moisture, or aerosol optical depth were assessed using the bootstrap method[87]. Specifically, we generated 1000 bootstrap samples through random samples with replacements from the original data to create samples of equal size to the original dataset. For each bootstrap sample, we computed the partial correlation coefficient between DTR and an environmental factor while adjusting for the potential confounding effect of other variables. Subsequently, we calculated the mean correlation coefficient across all bootstrap samples and determined the 95% confidence interval using the 2.5th to 97.5th percentile of the bootstrap distribution.

## Data availability

All data needed to evaluate the conclusions in the paper are present in the paper and/or the Supplementary Materials. The source data underlying Figs. 1–4 are provided as Source Data files and have been deposited in the Figshare repository available at https://doi.org/10.6084/m9.figshare.24310699.v1[88]. The CRU temperature dataset is from https://crudata.uea.ac.uk/cru/data/hrg/. The Berkeley Earth Surface Temperature dataset is from https://berkeleyearth.org/data/. The Global Surface Summary of the Day dataset is from https://data.nodc.noaa.gov/cgi-bin/iso?id=gov.noaa.ncdc:C00516. The ERA5 cloud cover, incident shortwave radiation and temperature dataset is from https://cds.climate.copernicus.eu/cdsapp#!/dataset/reanalysis-era5-single-levels-monthly-means?tab=form. The MODIS cloud dataset is from https://modis.gsfc.nasa.gov/data/. The Cloud and the Earth's Radiant Energy System energy balanced and filled cloud cover and incident shortwave radiation dataset is from https://asdc.larc.nasa.gov/project/CERES/CERES_EBAF_Edition4.1. The MERRA-2 downwards shortwave radiation, aerosol optical depth and surface albedo dataset

is from https://disc.gsfc.nasa.gov/datasets?project=MERRA-2. The GLEAM soil moisture data is from https://www.gleam.eu/#datasets.

## Code availability

The code for the analysis and mapping can be obtained from the Figshare repository (https://doi.org/10.6084/m9.figshare.24310699.v1)[88].

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

## Acknowledgements

This work has been supported by the Third Xinjiang Scientific Expedition Program (GrantNo.2022xjkk0106) and the State Key Laboratory of Earth Surface Processes and Resource Ecology. H.W.C. was supported by the Swedish Foundation for International Cooperation in Research and Higher Education (CH2020-8799).

## Author contributions

B.H. and Z.Z. designed research; Z.Z. performed the analysis and wrote the paper; H.W.C., D.C., and T.Z. provided comments to improve the manuscript; B.H. supervised the project. W.D., C.X., S.X., X.S., L.G., R.D., L.Z., L.H., W.Y., X.H., D.J., and X.Z. offered thoughts on the analysis and contributed to the writing.

## Competing interests

The authors declare no competing interests.
