## [Peer Review File · Nature Communications]

Reversed asymmetric warming of sub-diurnal temperature over land during recent decadesREVIEWER COMMENTS

Reviewer #1 (Remarks to the Author):

Review of "Reversed asymmetric warming of sub-diurnal temperature over land during recent decades by Zhong et al.

The paper documents a shift in the trends from the established diurnal asymmetric warming linked to earlier found processes that makes the climate change more efficient of increasing the nighttime temperatures than during daytime. The currently presented analysis shows a shift in that behavior, i.e. toward processes that increase the maximum temperature more than earlier, thus a reversed asymmetric warming and even an overall increase in the diurnal temperature range (DTR). Such a result is significant enough to merit publication in a high-profile journal.

The authors attribute the change in maximum temperature to a widespread decline in cloud cover, decreased aerosol optical depth in Western Europe and decreased soil moisture in southern South America. Here the story gets a bit complicated as the observations of these quantities are not as well established as the maximum and minimum temperatures, which are not as well established as the mean temperature. The reliability of the datasets and other possible processes to attribute the change to is not discussed in the manuscript, although some similar analyses of alternative datasets are presented. Earlier published datasets have large white areas and other issues and crucial references discussing this are missing. I propose that the authors reduce the number of datasets used, discuss the uncertainties in a robust way and sort out what reliably be concluded from the data. These aspects need to be improved before considering publication.

Comments

Page 1, Line 54: While I agree that the maximum and minimum temperatures provide useful information, care is needed as it is much harder to produce a gridded data set of these fields. In a state-of-the-art dataset that I used five years ago, several land areas were not even reported due to these uncertainties, such as most of S. America and Africa. What is the quality of these datasets now? How does the observational coverage vary over the studied periods? Reading in AR6 there is a discussion on the quality and reference to a study by Thorne et al. (2016a,b) that I do not see referenced here. It is also evident from Fig 1 that any regional conclusion are sensitive to the quality of the original data which varies substantially (would be good for the reader if the data source and period was clear in the figure and not only in the caption).

Page 2, Line 60: Here you list a number of factors, however, you mix physical parameters with manmade changes which is confusing. I miss surface albedo for example, or do you consider that to be in the land use change? The land use change covers more things though and I think that albedo might be better observed than soil moisture which I assume is supposed to cover one aspect of the drought you discuss below but changes in albedo are more widespread. It is also clear that the factors vary regionally and seasonally.

Page 6, Line 138: The information on the analysis here is very limited and quite cryptical as the reader

does not know which data that is used for the total cloud cover and what is a ridge regression and why do you use that?

Page 6, Line 140: Where is the total cloud cover from? Is that the best variable to use?

Page 7, Line 145: This has been seen in earlier studies, please refer to them.

Page 7, Line 154: What type of cloud data goes into the CRU TS data, is it synop data? Can you be sure that there is no time dependent observational bias in the datasets? The only one covering the earlier period is CRU TS. Is it independent or does it make use of the satellite data for the later period? Would be good to have difference plots in Fig S3.

Page 8, Figure 3. In some of your figures, North Africa is grey which I assume is of lack of data but it is not commented and not clear which dataset that is lacking information and why.

Page 8, Line 173: While I understand it is interesting to try to see the role of different types of clouds, but using cloud cover is a blunt method and using reanalysis, where the observational material varies over time, is tricky. Do we really trust the very strong trend in low-level clouds? Did you try to do the whole analysis only in the reanalysis world? That would be interesting to see if it shows the change in DTR trends and the relationship with clouds, soil moisture and aerosols.

Page 9, Figure 4: What is the meaning of the dots?

Page 10, Line 206: What does “controlling for PDSI or soil moisture” mean?

Page 10, Line 208: We already know that surface solar radiation influences the DTR!

Page 10, Line 224: Do you mean that the surface of the air cools?

Page 10, Line 227: The sentence starting with “However”, states that there should be a limited role for the drought and aerosols, that is not reflected in the abstract which claims that these two factors has a role. The only conclusion is that the solar radiation has changed due to changes in clouds and that results in a changed trend for DTR.

There are a number of issues in the Methods section, please rewrite and make sure that everything is logical and refer to the right dataset and explain all symbols and do not introduce symbols that are not used.

I will not comment any details in the supplement at this stage.

References

Thorne, P.W. et al., 2016a: Reassessing changes in diurnal temperature range: Intercomparison and evaluation of existing global data set estimates (2016b). *Journal of Geophysical Research: Atmospheres*, 121(10), 5138–5158, doi:10.1002/2015jd024584.

Thorne, P.W. et al., 2016b: Reassessing changes in diurnal temperature range: A new data set and characterization of data biases (2016a). *Journal of Geophysical Research: Atmospheres*, 121(10), 5115–5137,

Reviewer #2 (Remarks to the Author):

The manuscript considers how the diurnal temperature range (DTR) has increased during the last 30 years, in contrast to the previous 30 years where the DTR decreased. The authors explain this reversal in the DTR trend by comparing the contribution due to aerosols, cloud cover and drought, and suggest that the trend is primarily driven by changes in cloud cover and therefore incoming solar radiation at the surface, which increases T_{max} more than T_{min} .

The paper addresses an important trend and the results appear robust, even given various different datasets. However, there are some aspects of the statistical methods that are not explained in sufficient detail and I have two main concerns. Firstly, I expect more justification for the choice of regression and the features used. Secondly, the ridge regression model used frequently throughout the paper is not validated and I find it difficult to trust the results without evidence that ridge regression is a suitable and accurate model. Assuming this to be true, the remainder of the statistical analysis on the regression coefficients is valid and robust. The authors carefully address each potential driver of DTR changes and ensure robustness by repeating analyses on different datasets. The paper is clearly written and well presented. If the authors can show that the regression model is 1) justified by previous studies and 2) accurate in terms of prediction, I expect this paper to be an interesting contribution to the community.

Main Comments

L137: To explore

138 possible factors behind the changes in DTR in recent decades, we performed a ridge
139 regression analysis⁴⁹, reducing the impact of multicollinearity on the estimated
140 relationships during 1981–2020. ...

Further explanation of this method should be included here. Are the independent variables defined on every single grid cell? Why does one expect multi-collinearity? The dependent variable is DTR, is this a single value such as the global mean DTR? Or is this the DTR at every grid point?

L135-144

Before showing the values of the ridge regression coefficients, I think it is very important that the ridge regression model is validated. This can be done by leaving aside some samples from the training dataset and performing testing on these samples. Metrics such as R^2 , RMSE, etc. should be used to ensure the prediction from the ridge regression matches the test data. Metrics or plots highlighting the accuracy of this method should be included in the Supplementary. Then the full dataset can be used for the ridge regression in the remainder of the paper.

Limitations to the choice of model?

Are there any possible limitations to this study? I think there should be more justification for the model choices made in the introduction section, as well as comments on any limitations towards the end of the paper.

The paper focuses on aerosols, clouds and drought, which I understand are likely to be the dominant the dominant drivers of temperature change, but I would like to see more justification. Have previous studies confirmed that these are the main three drivers? I would also expect there to be additional contributors towards surface temperature changes that are not commented on. Can we be entirely confident that the regression does not ignore other important variables? What additional variables could be explored or probed in future studies?

Also, the ridge regression model used is a linear model. The authors do not comment on this assumption. Can this be justified based on previous studies?

Fig. 3.

This is a very interesting composite plot that nicely summarizes the coefficients. I have one query which would affect the output of the plot. Is the composite estimated based on the relative contribution from all three coefficients? i.e.

$$R = |\text{coefficient}_1| / (\text{coefficient}_1 + \text{coefficient}_2 + \text{coefficient}_3)$$

$$G = |\text{coefficient}_2| / (\text{coefficient}_1 + \text{coefficient}_2 + \text{coefficient}_3)$$

$$B = |\text{coefficient}_3| / (\text{coefficient}_1 + \text{coefficient}_2 + \text{coefficient}_3)$$

If so, this would mean that grid cells which have low coefficient values everywhere would still appear with equal “amount” of color. I highlight this because I notice that areas that have low magnitude values for all three coefficients in Fig S2. e.g. Southern Western Africa, parts of Canada, Siberia and Brazil, are still “bright” in Fig 3. They appear a little noisy. This could be misleading. Could the authors clarify the method for constructing the composite?

I think the confusion may come from the mention of “absolute” in the caption which states “The colour of the composite was determined by the absolute value of the ridge 172 regression coefficients”, given there are two possible meanings of absolute (“magnitude” or “absolute” vs. relative). For clarity this could be reworded, e.g. “The colour of the composite was determined by the relative contribution from the magnitude of the ridge regression coefficients”.

Methods

In some cases, the description of the statistical methods are limited. I would expect some more detail on the following:

1. Ridge regression: how is the penalization term added (L2 norm)? How is the penalization term tuned? The equation would be beneficial to readers here. This would also help clarify what the independent and dependent variables are and how many there are (i.e. global mean or individual grid cells?)
2. Partial correlation: this is not explained much for those who are not familiar with it and to clarify any choices made. My main questions are: how do you “control” for PDSI? Do you take it to be the mean value everywhere?

Also references could be added to the methods for those who would like further details.

Minor comments, grammar, typos

L86: “The change in global average Tmax reached the warming rate in
87 Tmin in recent decades, with an earlier surpassing moment in BEST than CRU TS”
Should this be “the warming rate in Tmax” rather than “the change in global average Tmax” or are these
equivalent?

L87 In the last 30-year window (1991–2020), both two datasets
88 show a stronger increase in the
89 global average of Tmax than that of Tmin.
This is not clear to me for CRU TS in Fig. 1a, it appears they have a the same increase

L133 – “All these evidences indicate” -> All this evidence indicates

L173 – which dataset is used for the low, medium and high-level cloud cover?

L276 – “resolution. on a 0.25° grid” remove period.

Fig. S1 – are these global temperature trends?

Fig. S13, 14, etc. – “The black stripes mark” dots rather than stripes.

Reviewer #3 (Remarks to the Author):

The authors present analysis of the changes in the diurnal temperature range from two station-based gridded datasets of surface air temperature (BEST and CRU). They find that there is a clear change in the late 1980s from a period where the DTR was decreasing due to the more rapid increase in diurnal minimum temperatures than maximum temperatures, to a period where either they are both increasing at the same rate or the diurnal maximum temperatures have been increasing more than diurnal minimum temperatures. Changes to the diurnal temperature range are extremely important to a range of impacts, and so this kind of analysis is very important for many groups. While the analysis seems reasonable and supports the conclusions I have a major concern about the inconsistency in the DTR changes between the CRU and BEST datasets which would need to be addressed prior to publication.

Major concern:

There is a large difference in the results from CRU and BEST (Figure 1) so why not also apply the rest of the analysis to BEST as well? I don't agree with the reasoning to focus on CRU as the change is more pronounced in BEST since the spatial patterns and hence many of the conclusions that follow in the rest of the paper are going to depend on which dataset is used. This large systematic uncertainty in changes

in the DTR needs to be addressed to check for consistency in the findings prior to publication.

Short comments:

Why are you showing the non-significant trends in Figure 2? Can't be sure these are increases/decreases rather than noise. I think it's more appropriate to put these together as non-significant trends to give a fuller picture of the surface area where we are seeing significant changes.

Response to reviewer: 1

The authors attribute the change in maximum temperature to a widespread decline in cloud cover, decreased aerosol optical depth in Western Europe and decreased soil moisture in southern South America. Here the story gets a bit complicated as the observations of these quantities are not as well established as the maximum and minimum temperatures, which are not as well established as the mean temperature. The reliability of the datasets and other possible processes to attribute the change to is not discussed in the manuscript, although some similar analyses of alternative datasets are presented. Earlier published datasets have large white areas and other issues and crucial references discussing this are missing. I propose that the authors reduce the number of datasets used, discuss the uncertainties in a robust way and sort out what reliably be concluded from the data. These aspects need to be improved before considering publication.

Res: Thanks for your constructive suggestions. In the revised manuscript, we have addressed the above-mentioned points as follows:

- (1) We have provided a more comprehensive discussion regarding the uncertainties associated with the data and the overall findings of the study.
- (2) We have reduced the number of datasets used to enhance clarity and reliability.
- (3) The issue of spatial gaps in some datasets is acknowledged and explained in the revised manuscript.

Moreover, we have made major revisions to the Methods section to provide more methodological details and clarify several points raised by you and the other reviewers.

More details can be found in our point-by-point responses below.

Page 1, Line 54: While I agree that the maximum and minimum temperatures provide useful information, care is needed as it is much harder to produce a gridded data set of these fields. In a state-of-the-art dataset that I used five years ago, several land areas were not even reported due to these uncertainties, such as most of S. America and Africa. What is the quality of these datasets now? How does the observational coverage vary over the studied periods? Reading in AR6 there is a discussion on the quality and reference to a study by Thorne et al. (2016a,b) that I do not see referenced here. It is also evident from Fig 1 that any regional conclusion are sensitive to the quality of the original data which varies substantially (would be good for the reader if the data source and period was clear in the figure and not only in the caption).

Res: You raised an important point regarding the difficulties in producing gridded datasets for maximum and minimum temperatures. We agree that there are uncertainties and challenges in obtaining complete coverage, especially in regions like South America and Africa where observational coverage remains limited. In our study, we used state-of-the-art datasets available at the time, but it is still worth noting that the quality and coverage of these datasets vary over the studied periods.

In the revised manuscript, we have discussed the limited observational coverage in South America and Africa, specifically referencing the study by Thorne et al. (2016a,b) for further context. Please see lines 271-286.

Moreover, we have conducted additional analyses, which revealed that even after excluding regions with higher uncertainties (most of South America and Africa), the average warming rate of maximum temperatures over the past three decades remains higher than that of minimum temperatures. Please see Fig. R1.

Fig. R1. Area-weighted average warming rates derived from the CRU TS (a) and BEST (b) datasets in T_{max} (red) and T_{min} (blue). The trends were calculated using a 30-year moving window over 1961–2020. The region highlighted in red in the insets delineate the study area here.

Furthermore, we have revised Fig. 1 following your suggestion to clearly indicate the data source and period in the figure itself, please see Fig. 1 in the revised manuscript.

Page 2, Line 60: Here you list a number of factors, however, you mix physical parameters with manmade changes which is confusing. I miss surface albedo for example, or do you consider that to be in the land use change? The land use change covers more things though and I think that albedo might be better observed than soil moisture which I assume is supposed to cover one aspect of the drought you discuss below but changes in albedo are more widespread. It is also clear that the factors vary regionally and seasonally.

Res: Changes in the physical parameters that were included can be manmade and natural. This study does not go into the causes of these changes. Regarding your question about the influence of surface albedo and soil moisture on DTR, we acknowledge that the original analysis overlooked the potential influence of surface albedo changes on DTR. Therefore, we have conducted additional partial correlation analyses in the revised manuscript, as shown in Fig. R2 below. Fig. R2a shows the spatial distribution of partial correlation coefficients between soil moisture and DTR,

controlling for the effects of cloud cover, aerosols, and surface albedo. Further, Fig. R2b shows the partial correlation coefficients between surface albedo and DTR, controlling for the effects of cloud cover, aerosols, and soil moisture. Overall, we found a significant negative correlation between soil moisture and DTR on a global scale, while the correlation between surface albedo and DTR was relatively weak and varied in different regions. A possible reason for this relationship between surface albedo and DTR is that a decrease in surface albedo can lead to increased absorption of solar radiation and sensible heat flux at the surface, resulting in higher maximum temperatures and DTR. Meanwhile, a decrease in surface albedo is often associated with an increase in vegetation greenness. The cooling effect of vegetation evapotranspiration can reduce maximum temperatures and DTR, resulting in the observed regional variations in the correlation between surface albedo and DTR. We have included a detailed discussion on this topic in the main text, as mentioned in lines 259-265 and 301-308.

Fig. R2. Spatial distribution of partial correlation coefficient between yearly diurnal temperature range (DTR) and soil moisture (a) or surface albedo (b) during 1981-2020. The soil moisture data was obtained from the Global Land Evaporation Amsterdam Model dataset and the surface albedo data was obtained from MERRA-2. The black dots mark the areas where correlations are significant at the $p < 0.05$ level.

Page 6, Line 138: The information on the analysis here is very limited and quite cryptical as the reader does not know which data that is used for the total cloud cover and what is a ridge regression and why do you use that?

Res: Sorry that the source of information for clouds were not provided. The total cloud cover data used in our study is derived from the ERA5 dataset, which is now clarified in lines 152-157 in the revised manuscript.

As for the use of ridge regression, it is a statistical method employed to reduce the impact of multi-collinearity among independent variables that influence the variation in DTR. Given the presence of multi-collinearity among the explanatory variables we considered, we deemed it necessary to use ridge regression to mitigate the impact of this issue. We have incorporated an explanation of ridge regression in the revised manuscript, see lines 140-161 and the expanded “Ridge regression” subsection in Methods.

Page 6, Line 140: Where is the total cloud cover from? It that the best variable to use?

Res: The total cloud cover data is sourced from the ERA5 dataset. Among the commonly used reanalysis datasets for cloud cover, only two datasets cover the period from 1961 to 2020: ERA5 and CRU TS. We chose the ERA5 dataset because the generation process of the cloud cover dataset in CRU TS involves the use of DTR. Specifically, DTR station anomalies were used to estimate ‘synthetic’ CLD station anomalies since 2003, by a linear transformation with a scaling factor and mean offset calculated from CRU TS gridded CLD and DTR values for each latitude band (Harris et al. 2014). Therefore, to ensure the integrity of the main analysis on the relationship between DTR and cloud cover, we did not use the CRU TS dataset. Consequently, in the main text of the manuscript, we rely on the cloud cover data

from the ERA5 dataset for our primary analysis, which is now mentioned in lines 152-154.

In addition to total cloud cover, the diurnal temperature range is also directly related to surface incoming radiation, as we will discuss in later sections. However, since the magnitude of surface incoming radiation is predominantly determined by total cloud cover, we have decided not to include both variables as independent variables in this analysis.

Reference cited:

Harris, I., Jones, P.D., Osborn, T.J. and Lister, D.H. (2014), Updated high-resolution grids of monthly climatic observations – the CRU TS3.10 Dataset. *Int. J. Climatol.*, 34: 623-642.
<https://doi.org/10.1002/joc.3711>

Page 7, Line 145: This has been seen in earlier studies, please refer to them.

Res: Point taken. We have cited relevant references at the specified location in the revised manuscript.

Page 7, Line 154: What type of cloud data goes into the CRU TS data, is it synop data? Can you be sure that there is no time dependent observational bias in the datasets? The only one covering the earlier period is CRU TS. Is it independent or does it make use of the satellite data for the later period? Would be good to have difference plots in Fig S3.

Res: The cloud cover data in CRU TS is a fused product derived from daily sunshine duration and diurnal temperature range observations, using empirical formulas. However, we cannot guarantee the absence of time-dependent observational bias during the data collection process. Therefore, we avoided using the cloud cover data from the CRU TS dataset.

Regarding the availability of cloud data covering the earlier period, our study primarily relies on the ERA5 reanalysis dataset, which covers the period from 1961 to 2020. The ERA5 dataset assimilates multiple satellite data sources and observational records. Therefore, we chose to analyze the ERA5 dataset due to its comprehensive coverage and assimilation of various data sources.

Page 8, Figure 3. In some of your figures, North Africa is grey which I assume is of lack of data but it is not commented and not clear which dataset that is lacking information and why.

Res: Sorry for not making this clear. The gray color in the North Africa region is due to the absence of PDSI data specifically in the Sahara region. In the revised manuscript, we have included an explicit explanation in the figure caption to clarify the reason for the missing information.

Page 8, Line 173: While I understand it is interesting to try to see the role of different types of clouds, but using cloud cover is a blunt method and using reanalysis, where the observational material varies over time, is tricky. Do we really trust the very strong trend in low-level clouds? Did you try to do the whole analysis only in the reanalysis world? That would be interesting to see if it shows the change in DTR trends and the relationship with clouds, soil moisture and aerosols.

Res: As you rightly pointed out, the trends in high, medium, and low-level cloud cover have a considerable amount of uncertainty, and we cannot fully trust the strong downward trend in low-level clouds in the reanalysis dataset. Therefore, our revised analysis is focused on total cloud cover. The issue of inhomogeneity (and thus

quality) of reanalysis is relevant. But on a positive note, the latest IPCC report (Chen et al., 2021) has assessed this issue and the improvement of reanalysis over the last years and its usefulness for climate change studies have been highlighted. Thus, we have some confidence in the ERA5 data. Based on your advice, we have revised this section to analyze the global long-term trends of DTR, cloud cover, soil moisture, and aerosols, as well as their relationships using reanalysis data only. Please see Fig. S6 and lines 198-212 in the revised manuscript.

Reference cited:

Chen, D., M. Rojas, B.H. Samset, K. Cobb, A. Diongue Niang, P. Edwards, S. Emori, S.H. Faria, E. Hawkins, P. Hope, P. Huybrechts, M. Meinshausen, S.K. Mustafa, G.-K. Plattner, and A.-M. Tréguier, 2021: Framing, Context, and Methods. In *Climate Change 2021: The Physical Science Basis. Contribution of Working Group I to the Sixth Assessment Report of the Intergovernmental Panel on Climate Change* [Masson-Delmotte, V., P. Zhai, A. Pirani, S.L. Connors, C. Péan, S. Berger, N. Caud, Y. Chen, L. Goldfarb, M.I. Gomis, M. Huang, K. Leitzell, E. Lonnoy, J.B.R. Matthews, T.K. Maycock, T. Waterfield, O. Yelekçi, R. Yu, and B. Zhou (eds.)]. Cambridge University Press, Cambridge, United Kingdom and New York, NY, USA, pp. 147–286, doi:10.1017/9781009157896.003.

Page 9, Figure 4: What is the meaning of the dots?

Res: Sorry for having missed the explanation. The black dots represent the regions where the trends reached a level of significance of $p < 0.05$. We have clarified this in the caption of Fig. 4 in the revised manuscript.

Page 10, Line 206: What does “controlling for PDSI or soil moisture” mean?

Res: It means that in the partial correlation analysis, we considered and adjusted for the influence of either PDSI or soil moisture on the relationship between DTR and solar radiation. We have revised the imprecise wording in the revised manuscript.

Page 10, Line 208: We already know that surface solar radiation influences the DTR!

Res: Point taken. We have modified the inaccurate wording and provided a more precise statement regarding the influence of surface solar radiation on DTR. Additionally, we have included relevant citations in the revised manuscript.

Page 10, Line 224: Do you mean that the surface of the air cools?

Res: Yes, we intended to convey the message that there is a cooling effect of soil moisture evaporation on air temperature. We have provided further clarification on this point in the revised manuscript.

Page 10, Line 227: The sentence starting with “However”, states that there should be a limited role for the drought and aerosols, that is not reflected in the abstract which claims that these two factors has a role. The only conclusion is that the solar radiation has changed due to changes in clouds and that results in a changed trend for DTR.

Res: In the revised manuscript, we have modified the expression in the abstract to ensure consistency with the statement in this section. Please refer to the revised abstract for the updated wording.

There are a number of issues in the Methods section, please rewrite and make sure that everything is logical and refer to the right dataset and explain all symbols and do not introduce symbols that are not used..

Res: Thank you for your constructive comments. We have thoroughly revised the Methods section to ensure logical flow, accurate referencing of datasets, and clear explanations of symbols. We have also removed any symbols that are not used in the study.

Response to reviewer: 2

The paper addresses an important trend and the results appear robust, even given various different datasets. However, there are some aspects of the statistical methods that are not explained in sufficient detail and I have two main concerns. Firstly, I expect more justification for the choice of regression and the features used. Secondly, the ridge regression model used frequently throughout the paper is not validated and I find it difficult to trust the results without evidence that ridge regression is a suitable and accurate model. Assuming this to be true, the remainder of the statistical analysis on the regression coefficients is valid and robust. The authors carefully address each potential driver of DTR changes and ensure robustness by repeating analyses on different datasets. The paper is clearly written and well presented. If the authors can show that the regression model is 1) justified by previous studies and 2) accurate in terms of prediction, I expect this paper to be an interesting contribution to the community.

Res: Thank you for your constructive and positive feedback. Following your suggestions, we have significantly expanded the justification and explanation for selecting ridge regression as our main analysis method. We have also conducted additional validation to demonstrate the accuracy of the ridge regression model in terms of prediction. Finally, we have conducted additional analyses using a Random Forest regression model to confirm that the linear assumption in the ridge regression is reasonable. These improvements have been incorporated into the revised manuscript, providing a more comprehensive understanding of the statistical methods used.

In addition to these new analyses, the revised manuscript also includes additional citations of previous studies to better support our choices of methods and findings.

L137: To explore

138 possible factors behind the changes in DTR in recent decades, we performed a ridge

*139 regression analysis⁴⁹, reducing the impact of multicollinearity on the estimated
140 relationships during 1981–2020. ...*

Further explanation of this method should be included here. Are the independent variables defined on every single grid cell? Why does one expect multi-collinearity? The dependent variable is DTR, is this a single value such as the global mean DTR? Or is this the DTR at every grid point?.

Res: The ridge regression analysis was indeed performed on a grid-based level, where both the independent variables and the dependent variable (DTR) were considered at individual grid points. This has been clarified in the revised manuscript.

As for the issue of multi-collinearity, we have discussed it in the specified paragraph of the revised manuscript at lines 142-150. Specifically, there is a strong correlation among the potential drivers of DTR, which include total cloud cover, aerosols, and soil moisture. For example, the phenomenon of aerosol-cloud interactions has received significant support from numerous published reports, with aerosols influencing cloud albedo and cloud lifetime. Additionally, clear associations exist between cloud cover and soil moisture. Increased cloud cover has the potential to enhance precipitation, leading to soil wetting. The complex interactions among these factors indicate a high degree of multicollinearity when conducting regression analysis, making it challenging to identify the dominant drivers of DTR changes.

L135-144

Before showing the values of the ridge regression coefficients, I think it is very important that the ridge regression model is validated. This can be done by leaving aside some samples from the training dataset and performing testing on these samples. Metrics such as R^2 , RMSE, etc. should be used to ensure the prediction from the ridge regression matches the test data. Metrics or plots highlighting the accuracy of this method should be included in the Supplementary. Then the full dataset can be used for the ridge regression in the remainder of the paper.

Res: We agree with you. Following your advice, we have partitioned all the data into an 80% training set and a 20% validation set to validate the ridge regression model. Meanwhile, the full dataset was employed to identify the ridge regression coefficients. Regions with dense distributions of temperature measurement stations, including North America, Europe, East Asia, and Australia, exhibit high R-squared values in the validation datasets. Please see Fig. S3a in the revised Supplementary Information.

Limitations to the choice of model?

Are there any possible limitations to this study? I think there should be more justification for the model choices made in the introduction section, as well as comments on any limitations towards the end of the paper?

Res: In the revised manuscript, we have included more justification for the choice of ridge regression analysis in the introduction section of the manuscript, specifically in the context of "Potential mechanisms behind reversing asymmetric warming". We explain why this modeling approach was employed and its relevance to our research objectives. Please see lines 140-161.

Additionally, we have addressed the limitations of the ridge regression model towards the end of the paper, discussing the uncertainties associated with its application. Please see lines 287-297.

The paper focuses on aerosols, clouds and drought, which I understand are likely to be the dominant the dominant drivers of temperature change, but I would like to see more justification. Have previous studies confirmed that these are the main three drivers? I would also expect there to be additional contributors towards surface temperature changes that are not commented on. Can we be entirely confident that the regression does not ignore other important variables? What additional variables could be explored or probed in future studies?

Res: While there is extensive literature suggesting that cloud cover, aerosol optical depth, and soil moisture can significantly influence DTR, they are not the exclusive drivers. In our revised manuscript, we have provided additional discussion on the complex influence of precipitation, atmospheric water vapor, albedo and land use/land cover changes (LULCC) on DTR, recognizing that it may be other important contributors to surface temperature changes that were not explicitly mentioned. We also suggested that the effect of LULCC on DTR could further be explored or probed to gain a more comprehensive understanding of DTR variability. Please see lines 294-308.

Also, the ridge regression model used is a linear model. The authors do not comment on this assumption. Can this be justified based on previous studies?

Res: As far as we know, there is no previous study that explicitly justified the linear assumption for this application. Thus, we have added an analysis using a Random

Forest regression model to test the robustness of our findings. The Random Forest regression model is capable of capturing complex, non-linear relationships. Our analysis using the Random Forest model yields results consistent with the ridge regression findings, which indicates that neglecting nonlinear relationships does not have a substantial impact on the research findings. This is discussed in the revised manuscript in lines 181-188.

Fig. 3.

This is a very interesting composite plot that nicely summarizes the coefficients. I have one query which would affect the output of the plot. Is the composite estimated based on the relative contribution from all three coefficients? i.e.

$$R = |\text{coefficient}_1| / (\text{coefficient}_1 + \text{coefficient}_2 + \text{coefficient}_3)$$

$$G = |\text{coefficient}_2| / (\text{coefficient}_1 + \text{coefficient}_2 + \text{coefficient}_3)$$

$$B = |\text{coefficient}_3| / (\text{coefficient}_1 + \text{coefficient}_2 + \text{coefficient}_3)$$

If so, this would mean that grid cells which have low coefficient values everywhere would still appear with equal “amount” of color. I highlight this because I notice that areas that have low magnitude values for all three coefficients in Fig S2. e.g. Southern Western Africa, parts of Canada, Siberia and Brazil, are still “bright” in Fig 3. They appear a little noisy. This could be misleading. Could the authors clarify the method for constructing the composite?

I think the confusion may come from the mention of “absolute” in the caption which states “The colour of the composite was determined by the absolute value of the ridge 172 regression coefficients”, given there are two possible meanings of absolute (“magnitude” or “absolute” vs. relative). For clarity this could be reworded, e.g.

“The colour of the composite was determined by the relative contribution from the magnitude of the ridge regression coefficients”?

Res: You have correctly understood our analysis process. In the revised manuscript, we have revised the corresponding description to avoid any potential misunderstanding. Please see lines 192-197. Additionally, we have provided further explanations in the Methods section to clarify the calculation process, please see lines 426-433. These revisions aim to improve the clarity and understanding of our methodology.

Methods

In some cases, the description of the statistical methods are limited. I would expect some more detail on the following:

1. Ridge regression: how is the penalization term added (L2 norm)? How is the penalization term tuned? The equation would be beneficial to readers here. This would also help clarify what the independent and dependent variables are and how many there are (i.e. global mean or individual grid cells?)

Res: Thank you for your valuable feedback on the statistical methods described in our manuscript. For each grid point in the regression analysis, the penalization term λ was initially set to 0 and incremented by a small step size of 0.01. As λ increased, multicollinearity decreased, leading to a corresponding decline in the Variance Inflation Factor (VIF) value. The incrementation of λ ceased when the VIF value dropped below 3, and this specific λ value was identified as the penalization term for the grid point. In the revised manuscript, we have provided a more comprehensive explanation of the penalization term, including the equation itself, to help readers better understand the methodology.

We have also clarified that the ridge regression analysis was performed at each individual grid cell, with cloud cover, aerosol optical depth, and soil moisture as the independent variables, and DTR as the dependent variable. These details are now elaborated upon in the revised Methods section. Please see lines 386-425.

2. Partial correlation: this is not explained much for those who are not familiar with it and to clarify any choices made. My main questions are: how do you “control” for PDSI? Do you take it to be the mean value everywhere?

Res: In our revised manuscript, we have provided a more comprehensive explanation of the partial correlation analysis, clarifying that it aims to assess the relationship between two variables while accounting for the influence of other variables. This technique helps isolate the associations between the variables of interest.

Regarding the control variable PDSI, we would like to clarify that we did not consider it as the mean value everywhere. Instead, we collected the specific values of solar radiation, DTR, and PDSI for each grid cell within our study area. These values were then used in the partial correlation analysis to evaluate the direct relationship between solar radiation and DTR while controlling for the potential confounding effects of PDSI.

Also references could be added to the methods for those who would like further details.

Res: We have followed your suggestion by adding appropriate references to the Methods section.

L86: *“The change in global average Tmax reached the warming rate in
87 Tmin in recent decades, with an earlier surpassing moment in BEST than CRU
TS”*

*Should this be “the warming rate in Tmax” rather than “the change in global
average Tmax” or are these equivalent?.*

Res: Yes, it should indeed be “the warming rate in Tmax”. We have corrected it in the revised manuscript.

L87 *In the last 30-year window (1991–2020), both two datasets
88 show a stronger increase in the
89 global average of Tmax than that of Tmin.*

*This is not clear to me for CRU TS in Fig. 1a, it appears they have a the same
increase.*

Res: We have revised the corresponding statement to make it clearer. Please see lines 88-90.

L133 – *“All these evidences indicate” -> All this evidence indicates.*

Res: Revised as suggested.

L173 – *which dataset is used for the low, medium and high-level cloud cover?*

Res: In the previous version of the manuscript, we used the ERA5 dataset for this analysis. However, based on the suggestion from Reviewer 1, in the revised version, we have replaced this section with an analysis of the annual variations of DTR and other environmental variables due to the large uncertainty in low, medium and high-level cloud cover in the reanalysis, especially in the earlier period.

L276 – “resolution. on a 0.25° grid” remove period.

Res: Fixed.

Fig. S1 – are these global temperature trends?

Res: Yes. We now mention this in Fig. S2 in the revised Supplementary Information.

Fig. S13, 14, etc. – “The black stripes mark” dots rather than stripes.

Res: Done.

Response to reviewer: 3

Major concern:

There is a large difference in the results from CRU and BEST (Figure 1) so why not also apply the rest of the analysis to BEST as well? I don't agree with the reasoning to focus on CRU as the change is more pronounced in BEST since the spatial patterns and hence many of the conclusions that follow in the rest of the paper are going to depend on which dataset is used. This large systematic uncertainty in changes in the DTR needs to be addressed to check for consistency in the findings prior to publication.

Res: We agree with you. The selection of DTR data needs to be handled with caution as it can significantly impact the subsequent analysis outcomes. In the revised manuscript, we have taken measures to address this concern. We validated both sets of DTR data against DTR observations and found that BEST data exhibited higher overall accuracy with the DTR observations. Consequently, we chose to use BEST data for the analysis of factors influencing DTR changes. Please see Fig. S1 and lines 121-127 in the revised manuscript. The conclusions drawn from this analysis align closely with those derived from the previous analysis using CRU data. Additionally, we have provided a comprehensive discussion on the uncertainties associated with both datasets. Please see line 267-286.

Short comments:

Why are you showing the non-significant trends in Figure 2? Can't be sure these are increases/decreases rather than noise. I think it's more appropriate to put these together as non-significant trends to give a fuller picture of the surface area where we are seeing significant changes?

Res: Good suggestion! We have revised the figure accordingly. Please see Fig. 2 in the revised manuscript.

REVIEWERS' COMMENTS

Reviewer #1 (Remarks to the Author):

The authors have done a good job on revising the manuscript based on my and the other reviewers' comments. I am mostly happy with the current version although I have some smaller suggestions for clarity.

Line 94: Would be interesting to know the fraction of the area that overlap between CRU TS and BEST. Is there a resolution or data treatment difference that makes out part of this difference? Seems like the CRU is much smoother field.

Lines 97 and 98: There is not enough confidence in the data to give the percentages with one decimal, please write 63 and 35 % instead.

Line 210: What about the influence of the clouds during nighttime? Less clouds during the night leads to more outgoing radiation and possibly lower minimum temperatures.

Line 240: It is kind of obvious that you found a stronger correlation between solar radiation during daytime, during nighttime there is no sunlight! However, there could be an influence on the nighttime temperatures that remain from the day before, but that is much harder to show and does not make much sense to go into that discussion here.

Line 287: Even though you have mentioned regression analysis I think you need to write regression models as it is easily misunderstood to be other types of models (e.g. climate models) is you only write models.

Lines 349 and 353: Why do you use two different resolutions of the ERA5? Or is it just two ways to write the same resolution? Please be consistent and remember that there is no exact translation of a lat-lon grid to km, it is not equal in north-south nor constant with latitude.

Figure 2: It is very difficult to distinguish between the small points with no trend and the blue dots. Please use a different color scheme. What does the * indicate, please provide information in the caption.

Figure 3: Please change the color scale to something which is possible for color blind to distinguish between, green and red is the worst combination. There are a number of websites that provide color pallets with enough contrast for everyone.

Figure 4: I think it is good if you mention the data source for each figure. At least in this one I am missing that information.

Reviewer #2 (Remarks to the Author):

The authors have done a nice job addressing my previous comments. I greatly appreciate the additional detail on ridge regression in both the main text and the methods. The newly added random forest analysis also strengthens the paper. The authors have also carefully outlined the limitations of the study in the new paragraph starting L289, which addresses many of my comments regarding limitations and choice of features. Aside from the following two very minor comments, I believe the manuscript is publication ready.

L406: I am not familiar with the meaning of “z-score anomalies”

L158 “Ridge regression is a linear regularization method ...”

To be more precise, I would refer to ridge regression as a regression method that uses regularization, rather than a regularization method in itself. I suggest something along the lines of “Ridge regression is a linear regression method with regularization ...”

Reviewer #3 (Remarks to the Author):

I am satisfied that the authors have addressed the concerns I raised.

Response to reviewer: 1

The authors have done a good job on revising the manuscript based on my and the other reviewers' comments. I am mostly happy with the current version although I have some smaller suggestions for clarity.

Res: Thank you, we very much appreciate your comments on how to further improve the manuscript and have revised the text accordingly, as detailed below.

Line 94: Would be interesting to know the fraction of the area that overlap between CRU TS and BEST. Is there a resolution or data treatment difference that makes out part of this difference? Seems like the CRU is much smoother field.

Res: Good suggestion! We have included additional information regarding the overlap of areas for both datasets. In response to your concern about the potential impact of resolution, we have conducted additional analyses. We resampled the CRU TS temperature data to a 1-degree latitude-longitude grid (matching the resolution of the BEST data) and recomputed the DTR, as shown in Fig. R1 below. The results are consistent with those obtained at the original resolution (0.5-degree latitude-longitude grid), indicating that resolution differences may not be a significant factor contributing to the variations between the two datasets. We believe that the reason why CRU is smoother is how the interpolation/extrapolation was done (Xu et al. 2018).

Xu, W., Li, Q., Jones, P. et al. A new integrated and homogenized global monthly land surface air temperature dataset for the period since 1900. *Clim Dyn* 50, 2513–2536 (2018).

<https://doi.org/10.1007/s00382-017-3755-1>

Fig. R1. Spatial distribution of the trend in DTR in CRU TS during 1961–1990 (a) and 1991–2020 (b). Here, the CRU TS temperature data was resampled to a 1-degree latitude-longitude grid.

Lines 97 and 98: There is not enough confidence in the data to give the percentages with one decimal, please write 63 and 35 % instead.

Res: We have made the suggested change accordingly.

Line 210: What about the influence of the clouds during nighttime? Less clouds during the night leads to more outgoing radiation and possibly lower minimum temperatures.

Res: Good question. We have included an analysis of the influence of cloud cover on both maximum and minimum temperatures, as shown in the new Fig. S7, which is also included below. Our findings indicate that, in some regions, there is indeed a positive correlation between cloud cover and minimum temperatures, likely due to the greenhouse effect of clouds that you mentioned. However, overall, a negative correlation between cloud cover and maximum temperatures appears to be more prevalent. This suggests that the effect of cloud cover on daytime solar incoming radiation generally plays a more important role in controlling the DTR. This is now mentioned on Lines 209–219.

Supplementary Fig 7. Spatial distribution of partial correlation coefficients between total cloud cover and maximum temperature (a) or minimum temperature (b) during 1981-2020, while controlling for the influence of soil moisture. The black dots mark the areas where correlations are significant at the $p < 0.05$ level. The total cloud cover was obtained from ERA5 dataset. (c) to (d), same as (a) to (b), but the total cloud cover was obtained from MODIS (2003-2020) dataset.

Line 240: *It is kind of obvious that you found a stronger correlation between solar radiation during daytime, during nighttime there is no sunlight! However, there could be an influence on the nighttime temperatures that remain from the day before, but that is much harder to show and does not make much sense to go into that discussion here.*

Res: Another effect is that the minimum temperature often occurs shortly after sunrise and thus could be affected by the amount of solar radiation around sunrise, but we agree that these effects are harder to show using our data sources and have therefore removed this part from the manuscript.

Line 287: *Even though you have mentioned regression analysis I think you need to write regression models as it is easily misunderstood to be other types of models (e.g. climate models) is you only write models.*

Res: The revised text now mentions ‘regression models’ instead of just ‘models.’

Lines 349 and 353: Why do you use two different resolutions of the ERA5? Or is it just two ways to write the same resolution? Please be consistent and remember that there is no exact translation of a lat-lon grid to km, it is not equal in north-south nor constant with latitude.

Res: The ERA5 data is produced and archive on a reduced Gaussian grid with a resolution of N320 and decreasing number of longitude points toward the poles. (Some upper air data is archived as spectral coefficients, but we will not go into these details here.) The native resolution of ERA5 is thus around 31 km. However, ERA5 data is commonly provided on a $0.25^\circ \times 0.25^\circ$ regular lat/lon grid (through bilinear interpolation for continuous data), which is also what we used here. Nevertheless, we think it is important to also point out the native resolution of the data. We agree that the previous text could cause confusion and have therefore revised it as follows:

“Monthly data on total cloud cover were obtained from the fifth-generation ECMWF reanalysis (ERA5) dataset⁵⁶ on a $0.25^\circ \times 0.25^\circ$ regular latitude–longitude grid (the native resolution of ERA5 is about 31 km).

...

The monthly incident shortwave radiation data in all-sky conditions were obtained from the ERA5 dataset on a 0.25° grid ...”

*Figure 2: It is very difficult to distinguish between the small points with no trend and the blue dots. Please use a different color scheme. What does the * indicate, please provide information in the caption.*

Res: Point taken. We have revised Figure 2 with a different color scheme to make it easier to distinguish between the small points with no trend and the blue dots. We have also updated the caption for improved clarity.

Figure 3: *Please change the color scale to something which is possible for color blind to distinguish between, green and red is the worst combination. There are a number of websites that provide color pallets with enough contrast for everyone.*

Res: Good suggestion! In the revised manuscript, we have replaced the red-green-blue (RGB) color scale with a cyan-magenta-yellow (CMY) composite plot, which enhances the figure's readability and ensures it is distinguishable for individuals with the most common color vision impairments. Please see the updated plots in Figure 3, Supplementary Figure 4, and Supplementary Figure 5.

Figure 4: *I think it is good if you mention the data source for each figure. At least in this one I am missing that information.*

Res: In the revised manuscript, we have included the data source in the caption for all figures.

Response to reviewer: 2

L406: I am not familiar with the meaning of “z-score anomalies”.

Res: Z-score are calculated by taking the difference between a data point and the mean of the dataset and then dividing it by the standard deviation. This standardization helps in comparing and interpreting data in a consistent manner, as it provides a measure of how unusual or extreme a data point is within the context of the dataset. We have added an explanation about z-score anomalies in the revised manuscript.

L158 “Ridge regression is a linear regularization method ...”

To be more precise, I would refer to ridge regression as a regression method that uses regularization, rather than a regularization method in itself. I suggest something along the lines of “Ridge regression is a linear regression method with regularization ...”

Res: We agree with you and have made the suggested change accordingly. Thank you for your suggestions!